# Identification of 3-Methoxyphenylacetic Acid as a Phytotoxin, Produced by *Rhizoctonia solani* AG-3 TB

**DOI:** 10.3390/molecules28020790

**Published:** 2023-01-12

**Authors:** Xinchun Li, HuiHui Hou, He Liu, Hancheng Wang, Liuti Cai, Mengnan An, Chong Zhang, Yuanhua Wu

**Affiliations:** 1Liaoning Key Laboratory of Plant Pathology, College of Plant Protection, Shenyang Agricultural University, Shenyang 110866, China; 2College of Economics and Management, Liaoning University of Technology, Jinzhou 121001, China; 3Guizhou Institute of Tobacco Science, Guiyang 556099, China

**Keywords:** *Rhizoctonia solani* AG-3 TB, 3-methoxyphenylacetic acid (3-MOPAA), phenylacetic acid (PAA), phytotoxin, toxin compound

## Abstract

Tobacco target spot disease is caused by *Rhizoctonia solani* AG-3 TB, which causes serious harm to the quality and yield of tobacco. In this study, thin layer chromatography (TLC), high performance liquid chromatography (HPLC), infrared absorption spectroscopy (IR), and nuclear magnetic resonance spectroscopy (NMR) were used to purify and identify the potential phytotoxin produced by *R. solani* AG-3 TB. The result indicated that the purified toxin compound was 3-methoxyphenylacetic acid (3-MOPAA) (molecular formula: C_9_H_10_O_3_). The exogenous purified compound 3-MOPAA was tested, and the results revealed that 3-MOPAA can cause necrosis in tobacco leaves. 3-MOPAA is a derivative of phenylacetic acid (PAA), which should be produced by specific enzymes, such as hydroxylase or methylase, in the presence of PAA. These results enrich the research on the pathogenic phytotoxins of *R. solani* and provide valuable insights into the pathogenic mechanism of AG-3 TB.

## 1. Introduction

*Rhizoctonia solani* Kühn is an important pathogenic fungus belonging to the soil-borne basidiomycete, and it can cause diseases in crops such as beans, tomatoes, potatoes, and tobacco [1,2]. *R. solani* can be classified into 14 different anastomosis groups, including AG-1 to AG-13 and a bridging isolate AG-BI, among which AG-1 IA can cause sheath blight in rice, AG-3 PT can cause black scurf in potato, and AG-3 TB can cause tobacco target spot disease [3]. Tobacco target spot disease was first recorded in the United States. Because of its gradual spread, tobacco fields in states such as California suffered heavy losses [4]. *R. solani* AG-3 TB infection in tobacco leaves results in yellow-brown spots with whorls, and it is highly destructive [2,5]. Owing to its rapid transmission, tobacco target spot disease was discovered in the Liaoning tobacco field in north-east China in 2012 [2]. Recently, this disease has occurred continually in Guangxi, Yunnan, Hunan, and Sichuan provinces, and the incidence area in the tobacco field is gradually expanding [6,7,8].

Pathogens can destroy host defenses by secreting toxins to damage plant tissue, producing effectors to suppress plant-cell death, and manipulating plant metabolism to favor infection by the pathogen [9]. Phytotoxins can be produced by pathogens, and most of them are secondary metabolites, which is a significant contributory factor in the expression of symptoms in host plants with a low molecular weight [10,11]. As one of the major pathogens, *R. solani* can reduce the crop yield and the quality of plants, causing damage to the stalks, leaves, and roots [12]. The toxin compounds secreted by *R. solani* were identified in as early as 1963. Carboxylic acid was first reported in the toxin compound of AG-1 IA, which includes succinic acid, phenylacetic acid (PAA), and furanic acid [13]. Subsequently, the phytotoxins, such as *m*-hydroxy- and *o*-hydroxy- phenylacetic acids, produced by *R. solani* AG-4 were isolated and identified, and these compounds were proven to cause the wilting of seedlings and other toxic effects [14]. In addition, 3-methylthiopropionic acid (3-MTPA) was isolated from *R. solani* AG-3 PT, and it was found to cause cell membrane and cytoplasm fracture [15]. Concerning toxin extraction refined by the Richard medium from *R. solani* AG-3 TB, it can damage the chlorophyll and inhibit the seed germination and radicle growth of tobacco [16]; however, the structures of the pathogenic toxins are still unclear.

Here, to clarify the special phytotoxin structure of *R. solani* AG-3 TB, we used thin-layer chromatography (TLC), high-performance liquid chromatography (HPLC), infrared absorption spectroscopy (IR), and nuclear magnetic resonance spectroscopy (NMR) to extract and identify the effective compound. The result revealed that 3-methoxyphenylacetic acid (3-MOPAA) as an important toxin compound was first identified in *R. solani* AG-3 TB. The exogenous purified compound 3-MOPAA can also cause the necrosis of tobacco leaves. We also propose that 3-MOPAA is a kind of derivative of PAA, which may be produced under certain special enzymes in the presence of PAA. The results provide a theoretical research basis for the phytotoxin chemical compound of *R. solani* AG-3 TB and elucidate the pathogenic mechanism of *R. solani*.

## 2. Results

### 2.1. Tissue Infection and Symptom Changes in R. solani AG-3 TB Infection

To clarify the changes of host symptoms during *R. solani* AG-3 TB infection, the characterizations were observed at 24, 48, and 72 h post inoculation (hpi). The result indicated that a yellow halo could be observed around the needle point at 24 hpi; at 48 hpi, the yellow halo had gradually expanded. At 72 hpi (Figure 1), the symptom around the needle point on the host was clear, with the appearance of a wheel-like pattern. The necrosis diameters around the needle point were 0.126 ± 0.087 cm, 0.340 ± 0.196 cm, and 0.606 ± 0.266 cm at 24 hpi, 48 hpi, and 72 hpi, respectively (Table 1).

### 2.2. Biological Activity, Isolation, and Purification of Toxin Compound of R. solani AG-3 TB

Some studies have reported that toxin extraction can inhibit radicle elongation and induce leaf necrosis and stem canker [15]. Here, the radicle elongation and lesion diameter of tobacco were determined by performing toxin extraction from *R. solani* AG-3 TB. The result demonstrated that the inhibition of radicle elongation was clear after toxin-extraction treatment. Additionally, the inhibition rate of the toxin extraction on radicle elongation was 93.14% (Appendix A). The lesion diameter of each phase of the toxin extracted by the activated carbon adsorption extraction method was different: the lesion diameter in the aqueous phase was 0.892 ± 0.165 cm, and the lesion diameter in the organic phase was 0.042 ± 0.018 cm (Appendix A). Therefore, this toxin extraction method was used to purify and identify the toxin compound in the following work. 

The toxin extraction was separated into four compounds with thin-layer chromatography (the ratio of layer-developing reagent was 95% ethanol: concentrated ammonia water: ethyl acetate = 10:1:1) (Figure 2A). Then, the pathogenicity of four compounds was determined. The result indicated that compound three had strong pathogenicity to tobacco leaves, and the lesion diameter was 0.9341 cm. Compound **4** had weak pathogenicity to tobacco leaves, while compounds **1** and **2** had no pathogenicity (Appendix A). 

Toxin compound **3** was detected using HPLC. The result indicated that seven compound peaks were detected, and the retention times were 1.62 s, 1.785 s, 2.043 s, 2.382 s, 3.366 s, 4.233 s, and 6.071 s, respectively (Figure 2B). The seven compound peaks were collected by preparative HPLC, and their biological activities were determined. The lesion diameter was 0.792 ± 0.273 cm after inoculation with compound V, which was collected using preparative HPLC (Appendix A). This toxin compound was detected using analytical HPLC, and a single peak was detected (Figure 2B).

### 2.3. Structure Elucidation of the Toxin Compound

The compound structure was determined using infrared absorption spectroscopy (IR) to clarify the structure of the toxin compound. In the IR spectrum, the O-H telescopic vibration, C=O telescopic vibration, C-O telescopic vibration, and O-H bending vibration were observed at 3400~2500 cm^−1^, 1698 cm^−1^, 1410 cm^−1^, and 1234 cm^−1^; these results indicate that carboxylic acid (- COOH) existed in this structure. According to the =C-H telescopic vibration, benzene ring skeleton C=C telescopic vibration, and =C-H out of plane bending vibration at 3017 cm^−1^, 1601 cm^−1^ and 1494 cm^−1^, 881 cm^−1^ and 793 cm^−1^ and 704 cm^−1^, 1,3-disubstituted benzene existed in this structure. At 2968 cm^−1^, 2921 cm^−1^, and 2841 cm^−1^ C-H telescopic vibration was observed, while at 1469 cm^−1^, and 1435 cm^−1^ a C-H bending vibration was observed; therefore, − CH_2_ − and − OCH_3_ were included in this compound. Additionally, the results of C-O-C telescopic vibration at 1267 cm^−1^, 900 cm^−1^, and 757 cm^−1^ illustrated that ArOR existed in this structure (Figure 3A, Table 2). According to the results of IR, carboxylic acid (COOH), 1,3-disubstituted benzene, aromatic ether (ArOR), − CH_2_ − and − OCH_3_ were contained in this toxin compound.

The mass-charge ratio of the [M-H]–peak of the toxin compound was 165.0 according to the result of mass spectrometry. This mass charge ratio of the toxin compound had a relative error with a theoretical value of the 3-MOPAA compound (−0.33 ppm). Therefore, the ionic molecular formula of this product may be C_9_H_9_O_3_. In addition, the results of an elementary analysis revealed that this compound contained C, H, O without a N atom. The ratio of C, H, O in the substance should, therefore, be 9:10:3. According to functional groups detected by infrared spectroscopy, the molecular formula of the toxin compound was inferred to be C_9_H_10_O_3_ (Figure 3B, Appendix A).

The structure of the toxin compound was deduced using nuclear magnetic resonance (NMR) (Figure 4A). The results of ^1^H-NMR indicated that six groups of peaks were produced. Their integral ratio (from low field to high field) was 1:1:1:2:3:2, and the total number of protons was 10, which was consistent with the proton number of 3-MOPAA. According to the chemical shift values (Appendix A) and the COSY and HMBC experiments (Figure 4C,D), the structure of the toxin was a spin system composed of four aromatic protons on 1,3-disubstituted benzene (δ 7.25 (1H, m), 6.87 (1H, d), and 6.83 (2H, m)), among which the δ 7.25 proton had a relation with two aromatic seasons C of δ 159.8 and 134.7 protons, and it was H5, while the chemical shifts of the other three protons were close, and they were H2, H4 and H6. The δ 3.80 (3H, s) and 3.62 (2H, s) values indicated two groups of isolated protons, which were H9 and H7, respectively. In addition, the values of δ 15~11 (1H, br) indicate carboxylic acid active protons -COOH.

According to the results of the carbon spectrum, displacement table and the HSQC two-dimensional spectrum (Figure 4B,E, Appendix A), δ 41.2 and 55.3 were two saturated Cs, namely C7 and C9, respectively. The results at δ 113.0, 115.2, 121.8, and 129.8 indicate four tert-aromatic Cs, among which δ 129.8 was C5, the other three were uncles Cs; furthermore, δ 113.0 represented C4, and δ 115.2 and 121.8 were C2 and C6. Three unsaturated C seasons were detected at δ 134.7, 159.8, and 178.0; on the basis of the displacement, they were C1, C3 and C8. Therefore, the structure of the toxin sample was consistent with the structure of 3-MOPAA (Figure 4A).

### 2.4. Pathogenicity and Virulence of Exogenous 3-MOPAA on N. tabacum

To clarify the pathogenicity and virulence of exogenous 3-MOPAA, the tobacco leaves were treated with 3-MOPAA at different concentrations (1 mg/mL, 2 mg/mL, 4 mg/mL). The results indicated that 3-MOPAA can cause necrosis around the inoculation point of leaves at 24 hpi (Figure 5). Additionally, the lesion diameter was 0.383 ± 0.0894 cm after inoculation with a treatment of 1 mg/mL of 3-MOPAA, while a lesion diameter of 0.654 ± 0.213 cm appeared after a 4 mg/mL treatment of 3-MOPAA (Table 3). 

## 3. Discussion

A total of 14 different subpopulations of *Rhizoctonia solani* were discovered. Because of its special multi-nuclear and multi-subgroup genetic characteristics, most of the research focused on the genetic diversity of *R. solani* or the resistance mechanism of host plants. Toxins, as one of the main pathogenic factors in the process of fungal infection, can affect physiological function, destroy cell structure, and cause metabolism disorder in plants at low concentrations [17,18]. Therefore, the structure and synthesis of phytotoxins have become important aspects in the management of fungal disease.

To further the research on *R. solani*, toxin compounds of AG-1 IA, AG-4, and AG-3 PT subgroups were isolated, including PAA, succinic acid, furanic acid, *o*-hydroxy-phenylacetic acid (*o*-OH-PAA), *m*-hydroxy- phenylacetic acid (*m*-OH-PAA), and 3-methylthiopropionic Acid (3-MTPA) [13,15,19,20]. In this study, 3-methoxyphenylacetic acid (3-MOPAA) (molecular formula: C_9_H_10_O_3_) was identified from *R. solani* AG-3 TB, which had the same structure when isolated from the AG-4 subgroup [14]. It is interesting that toxin compounds from *R. solani*, such as *o*-OH-PAA, *m*-OH-PAA, and 3-MOPAA, were derivatives of PAA [20]. PAA was first reported in plants that had similar functions to IAA; for example, the stimulation of callus formation and the promotion of plant growth, especially in lateral root induction [21,22,23,24]. Subsequently, studies reported that PAA can also be produced by microorganisms such as *Enterobacter cloacae*, *Bacillus subtilis* and *Staphylococcus aureus*, which are implicated in group irritability and biofilm destruction [25,26,27]. Moreover, PAA produced by fungi, including *Sporobolomycos roseus* and other species from the Asco-, Basidio-, Deutero-, and Phyco-mycetes, may have a close relationship with defense [28,29]. PAA as a toxin compound was first reported from the metabolites of *R. solani* in 1963 [13], and the genome prediction and transcriptome analysis of the *R. solani* AG-1 IA subgroup indicated that PAA may be produced under the five crucial enzymes, including: shikimate kinase, 3-phosphoshikimate 1-carboxyvinyltransferase (EPSP synthase), chorismate synthase, prephenate dehydrogenase, and prephenate dehydratase [30,31]. Other studies of the secondary metabolite gene from *R. solani* AG-3 TB also revealed that the gene expression level of critical enzymes in PAA synthesis increases during pathogen infection, especially in the early stage (6–12 hpi) and the middle stage (24–36 hpi) [32]. In addition, studies indicating that PAA produced by *R. solani* has harmful effects on the host are unavailable, but it has been found to play an important role in the growth of mycelium, while the PAA derivatives can cause the development of necrotic lesions in rice, rape, and potato [13,15,19,20,33]. Therefore, the compound 3-MOPAA isolated and identified from *R. solani* AG-3 TB belonged to the PAA derivatives, which may potentially be the source of valuable phytotoxin production.

The secondary metabolites produced by fungus are derived from central metabolic pathways and primary metabolite pools [34]. The different additional enzymes, such as synthetases, ligases, and dehydratases, were produced to ‘decorate’ the metabolites generated by the backbone enzymes, which can alter the bioactivities of the metabolites [34,35]. According to previous studies, a certain relationship between PAA and PAA derivatives confirmed that PAA derivatives can cause blocking-up by altering and limiting the production of PAA [20]. In this study, 3-MOPAA was revealed to be a potential phytotoxin in *R. solani* AG-3 TB. Additionally, PAA can be produced during *R. solani* AG-3 TB infection [32]. Therefore, some necessary connection must exist between PAA and 3-MOPAA, and we propose that PAA plays an important role in 3-MOPAA synthesis and that it is first produced by the critical five enzymes; then, 3-MOPAA is finally synthesized by special tailoring enzymes, such as hydroxylases and methyltransferases, during *R. solani* AG-3 TB infection. According to the phytotoxin research, we also consider that the phytotoxin (3-MOPAA) produced by AG-3 TB can be prevented by blocking the synthesis of PAA. Collectively, our results provide valuable insights into the phytotoxin compound *R. solani* AG-3 TB and validate its function during fungus infection.

## 4. Materials and Methods

### 4.1. R. solani AG-3 TB Isolation and Inoculation

The isolated and purified AG-3 TB strain (YC-9) was activated in potato dextrose agar (PDA) medium at 28 °C for 3 days. The PDA containing AG-3 TB (6 mm) was inoculated on the 5th and 6th tobacco leaves at the 9th leaf stage (Var.: NC89, a commonly cultivated susceptible variety). Cotton with sterile water was used for moisturizing, and the symptom was observed after inoculating the acupuncture part at 24, 48, and 72 hpi [32]. Four leaves of each tobacco plant were inoculated, and each leaf was inoculated at four acupuncture points.

### 4.2. Preparation and Extraction of Toxin from R. solani AG-3 TB

The high pathogenic strain YC-9 of *R. solani* AG-3 TB was transferred into PDA medium and cultured at 28 °C for 4 days. Each bottle was inoculated with 10 pieces of pathogen-containing PDA (5 mm in diameter) and cultured at 28 °C in the dark for 15 d; the bottle was shaken every 12 h. Three layers of 11 cm × 11 cm medium-speed qualitative filter paper and a vacuum suction bottle were used. The toxin extraction of *R. solani* AG-3 TB was achieved with activated carbon adsorption extraction [36]. The toxin extraction was evaporated to 1/10 of the original volume. Methanol (V:V = 1:1) was added, and it was left for adsorption for 12 h. Then, the sediment was removed, and 3% active carbon was added. The activated carbon was removed after 12 h, and 3 times the volume of ethyl acetate was added for extraction; the organic phase was rotated and evaporated to complete the process of toxin extraction from the pathogen. 

### 4.3. Biological Activity by Toxin Extraction 

The germinating seeds were picked out and placed in the culture dish covered with filter paper. The number of tobacco seeds (Var.: NC89) was 40, among which a total of 20 seeds were placed into a sterile-water culture dish, and other seeds were placed into an experience culture dish with 6 mL of toxin extraction (aqueous phase). The sterile water was used as CK, and the radicle length was measured.
Inhibition rate of radicle elongation (%) = 
$$\frac{\left(\mathrm{Average}\;\mathrm{length}\;\mathrm{of}\;\mathrm{control}\;\mathrm{radicle}-\mathrm{Average}\;\mathrm{length}\;\mathrm{of}\;\mathrm{treated}\;\mathrm{radicle}\right)}{\mathrm{Average}\;\mathrm{length}\;\mathrm{of}\;\mathrm{control}\;\mathrm{radicle}}\times 100$$

The toxin extract (15 μL) was inoculated on the healthy tobacco leaves (Var.: NC89) with sterilized needles; the culture solution was used as the control group, the inoculated tobacco leaves were cultured at 28 °C, and the diameter of the lesion was recorded 3–5 d after inoculation.

### 4.4. Isolation and Purification of Toxin Compound

The silicone–rubber plate (10 cm × 10 cm) was placed in a 60 °C oven for 15 min. One 1.5 cm straight line was drawn away from the plate and marked the spot at an interval of 1 cm. Each dot was spotted using a capillary pipette. The silica–gel plate was placed into the chromatography cylinder (with developing agent). The silica-gel plate was taken out for drying when the developing agent was 1 cm away from the plate. The Rf result was analyzed using the ultraviolet analyzer (λ = 254). Then, the compound was scraped off and collected for subsequent tests. 

Analytical HPLC was used to detect the purity of the toxin compound. The toxin compound was dissolved with methanol and filtered using a 0.22 μM water-filter membrane. The detection conditions were as follows: the flow rate was 1.0 mL/min; the injection volume was 20 μL; the detection wavelength was 275 nm. The pure sample was determined by the percentage of the measured peak area in the total peak area.

### 4.5. Structural Identification of Toxin

The sample was prepared using the KBr compression method. The 1.8 mg pure toxin and KBr was ground into powder and made into a diaphragm with a tablet machine. The infrared absorption peak was determined using a Nicolet 6700 Fourier transform spectrometer (Thermo fisher Company, Waltham, MA, USA). The scan number was set to 32, the resolution was 4 cm^−1^, and the scanning range was 400–4000 cm^−1^.

The mass charge ratio (*m*/*z*) was obtained by establishing the mass spectrum of the toxin sample using a combined quadrupole orbitrap mass spectrometer. The distilled water was used as a mobile phase solvent, the ESI (-) ionization mode was adopted, and the spray voltage was set at 2.5 KV. The toxin samples were detected using the German Elemental Verio MICRO (MICRO-EPSILON Precision Measurement Technology Co., Ltd.; Hanover, Bavaria, Germany) cube element analyzer. 

The toxin sample was dissolved with tritiated methanol, and then analyzed using a BRUKER-400 MHZ superconducting nuclear magnetic resonance instrument. The structure of the toxin was analyzed according to the results of ^1^H-NMR and ^13^C-NMR spectra.

### 4.6. The Pathogenicity of Purified Toxin Compound 3-MOPAA

Needle inoculation of detached leaves was performed. 3-MOPAA concentrations (dissolved with sterile water) at 1 mg/mL, 2 mg/mL, and 4 mg/mL were used for inoculation on tobacco leaves (NC89). Sterile water was used as the control and cultured at 28 °C for 24 h, and the diameter of the lesion was measured. Each treatment was tested 3 times, and the 3 leaves were inoculated as one biological replicate.

## Figures and Tables

**Figure 1 molecules-28-00790-f001:**
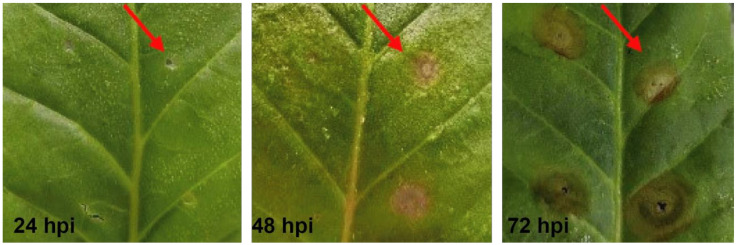
The symptoms of *Nicotiana tabacum* inoculated with *R. solani* AG-3 TB. Red arrows were symptoms of *R. solani* AG-3 TB infection.

**Figure 2 molecules-28-00790-f002:**
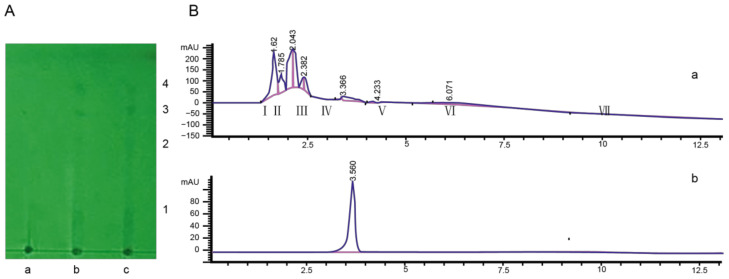
Isolation and purification of toxin components from the tobacco target spot pathogen. (**A**) The compounds of toxin filtrate were separated using TLC. The ratio of layer developing reagent was 95% ethanol: concentrated ammonia water: ethyl acetate = 10:1:1. a: blank filtrate; b and c: toxin filtrate. **1**, **2**, **3**, and **4** were different compounds. (**B**) The TLC compound **3** was tested using HPLC. a: compound **3** tested by HPLC; b: purity detection spectrum of toxin compound.

**Figure 3 molecules-28-00790-f003:**
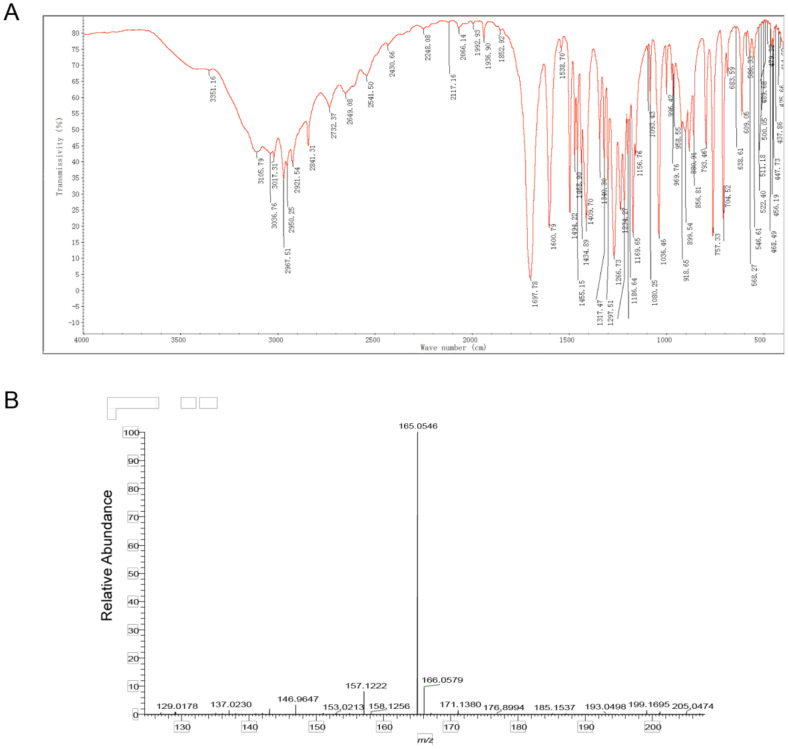
Identification of toxin compound by IR and mass spectrometry from *R. solani* AG-3 TB. (**A**) IR spectrum of toxin compound. (**B**) Mass spectrometry of toxin compound.

**Figure 4 molecules-28-00790-f004:**
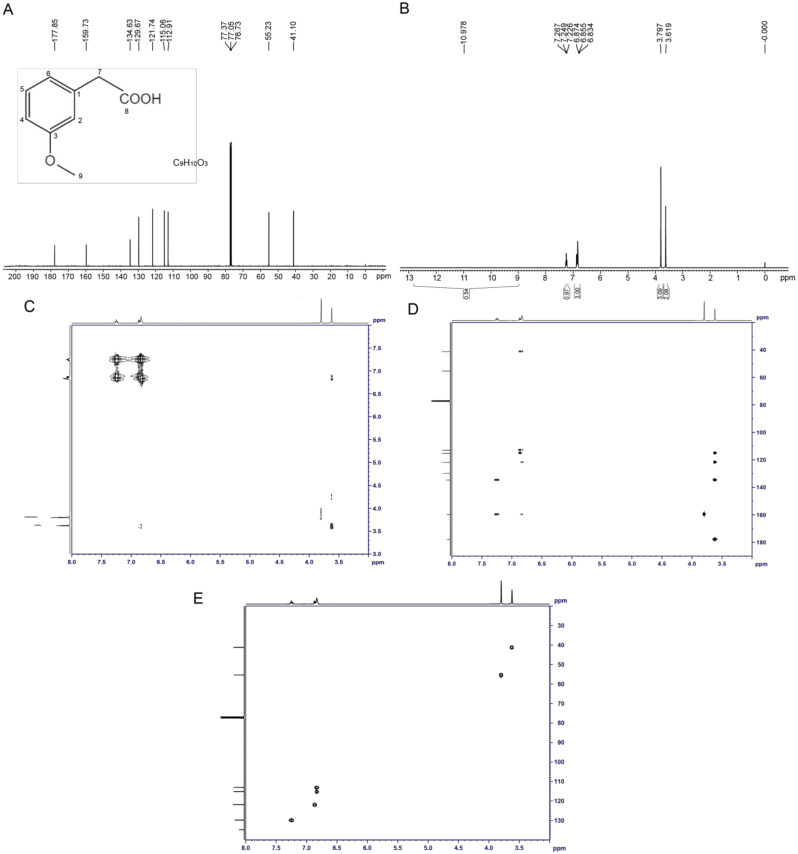
Identification of the structure of the toxin compound from *R. solani* AG-3 TB. (**A**) The ^1^H-NMR analysis of the structure of the toxin. (**B**) The ^13^C-NMR analysis of the structure of the toxin. (**C**) COSY NMR spectrum of the toxin compound. (**D**) HMBC NMR spectrum of the toxin compound. (**E**) HSQC NMR spectrum of the toxin compound.

**Figure 5 molecules-28-00790-f005:**
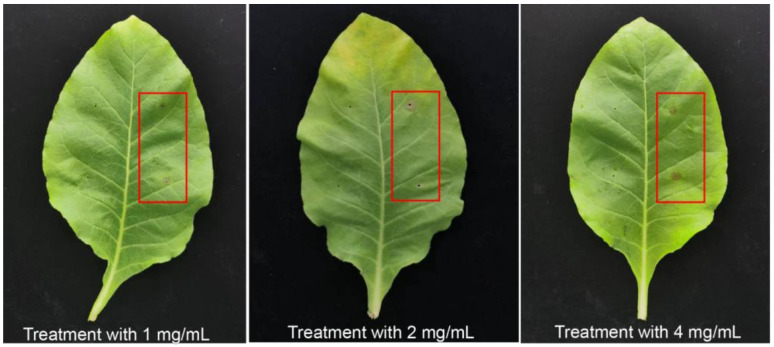
The symptoms of 3-MOPAA inoculation on *N. tabacum* (NC89) with different concentrations. Sterile water was used as control, and the red frames were the treatment groups with 3-MOPAA.

**Table 1 molecules-28-00790-t001:** The lesion diameter of needle point on *N. tabacum* inoculated with *R solani* AG-3 TB.

Hours Post Inoculation (hpi)	24 hpi	48 hpi	72 hpi
Lesion diameter (cm)	0.126 ± 0.087	0.340 ± 0.196	0.606 ± 0.266

**Table 2 molecules-28-00790-t002:** The element composition of toxin compound by IR.

Wave Number of Absorption Peak (cm^−1^)	Vibration Type	Group	Absorption Peak Intensity
3400~2500	O-H Telescopic vibration	Carboxylic acid (− COOH)	br
3017	=C-H Telescopic vibration	=C−H	m
2968, 2921, 2841	− C-H Telescopic vibration	Saturated − C-H	s, m, m
1698	C=O Telescopic vibration	Carboxylic acid (− COOH)	s
1601, 1494	Benzene ring skeleton C=C Telescopic vibration	Benzene ring	s, s
1469	C-H Bending vibration	− CH2 −	s
1435	C-H Bending vibration	− OCH3	s
1410	C-O Telescopic vibration	Carboxylic acid (− COOH)	s
1267, 900, 757	C-O-C Telescopic vibration	Aromatic ether (ArOR)	s, s, s
1234	O-H Bending vibration	Carboxylic acid (− COOH)	s
881, 793, 704	=C-H Out of plane bending vibration	1,3- disubstituted benzene	s, s, s

**Table 3 molecules-28-00790-t003:** The exogenous 3-MOPAA effect with different concentrations for lesion diameter.

Compound Treatment	1 mg/mL	2 mg/mL	4 mg/mL	CK
Lesion diameter (cm)	0.383 ± 0.0894	0.550 ± 0.171	0.654 ± 0.213	0.00

## Data Availability

The data can be found in Appendix A.

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
