# Peer review of "Identification of 3-Methoxyphenylacetic Acid as a Phytotoxin, Produced by Rhizoctonia solani AG-3 TB"

_molecules, 2023, doi:10.3390/molecules28020790_

Round 1
Reviewer 1 Report
General comments:
I have reviewed the manuscript entitled “Identification of 3-methoxyphenylacetic acid as a phytotoxin, produced by Rhizoctonia solani AG-3 TB” and my comments are as follows.
Overall, the manuscript is logical clear and easy to follow. However, the manuscript needs careful editing and particular attention to English grammar, spelling, and sentence structure. There are many things to be clarified.
All the marked places in the PDF version must be revised.
Specific comments:
1. Lines 29-33, delete “the” in these sentences.
2. Lines 38-40, I suggest to change the sentence to “Recently, the disease has been continuously occurred in Guangxi, Yunnan, Hunan, Sichuan provinces and ……”.
3. Lines 43-45, There are many phytotoxins in nature, so the sentence should be re-written.
4. Lines 51-52, I suggest to change the sentence to “and these compounds were proved to cause seedling wilting and other toxic effects.”.
5. Line 56, I suggest to revise the sentence to “however, the structures of the pathogenic toxins are still unclear.”.
6. The quality of Figure 2 is low, it should be re-prepared and replace it by the high resolution ones.
7. Lines 100-101, I don’t think the purity of the toxin can be simply determined by HPLC according to the number of peaks.
8. Line 155, “Figure 5A” is wrong.
9. Too many “the” were incorrectly used in the main text.
10. In section “4.2. Preparation and Extraction of toxin from R. solani AG-3TB”: The organic phase was rotated and evaporated to obtain the toxin extraction of pathogen (Lines 244-245). I strongly suggest the authors to clarify what kinds of solutions were used for toxin dissolution in radical elongation experiment (Lines 247-250), pathogenicity experiment (Lines 282-286).
11. Lines 282-286, I would like the authors to clarify how the concentrations of methanol extract of BS45 were determined in the manuscript.
12. I suggest the authors to provide the original data of the radical elongation experiment. Besides, according to the description by the authors I don’t know what kind of material was used in the radical elongation experiment.
13. Is the pathogenicity and virulence of the purified toxin, 3-MOPAA, consistent with the un-purified toxin extraction mix in the radical elongation experiment and pathogenicity experiment? I think further work should be conducted to confirm the conclusions.
14. All the references listed in the reference section should be double checked, for example the font case.
15. What’s the significance of the study? It’s better to emphasize the main contribution of this work.

Author Response
Dear Editor and Reviewers,
Thank you for your letter and the reviewers’ comments concerning our manuscript entitled “Identification of 3-methoxyphenylacetic acid as a phytotoxin, produced by Rhizoctonia solani AG-3 TB” (ID: molecules-2109950). Those comments are all very valuable and helpful for revising and improving our manuscript, as well as the important guiding for our researches. We have studied comments carefully and made correction in the manuscript. We have addressed almost of the queries raised by two reviewers, and the point-by-point response was shown in bellow. The revised part was marked with red in manuscript. Please see the attachment.

Reviewer 2 Report
Lines 2-3, change ‘Identification of 3-methoxyphenylacetic acid as a phytotoxin, produced’ to ‘Identification of 3-Methoxyphenylacetic Acid as a Phytotoxin, Produced’
Line 13, change ‘cause’ to ‘causes’
Line 14, delete ‘the’
Line 15, change ‘(IR)’ to ‘(IR), ’
Line 17, change ‘structure of’ to ‘produced by’
Line 19, delete ‘at different concentrations’
Line 29, change ‘potato’ to ‘potato,’
Line 33, change ‘The tobacco’ to ‘Tobacco’
Line 39, change ‘Yunnan, Hunan, Sichuan’ to ‘Yunnan, Hunan, and Sichuan’
Line 42, change ‘death and’ to ‘death, and’
Line 46, change ‘leaves and’ to ‘leaves, and’
Line 49, change ‘and furanic acid’ to ‘, and furanic acid’
Line 51, change ‘proved’ to ‘proven’
Line 57, delete ‘the’
Line 68, change ‘infection and symptom changes in R. solani AG-3 TB infection’ to ‘Infection and Symptom Changes in R. solani AG-3 TB Infection’
Line 70, change ‘48, 72 hours’ to ‘48, and 72 hours’
Line 75, change ‘48 hpi and 72 hpi’ to ‘48 hpi, and 72 hpi’
Line 78, delete ‘strain’
Line 79, delete ‘after’
Line 80, change ‘activity, isolation and purification of toxin compound’ to ‘Activity, Isolation, and Purification of Toxin Compound’
Line 97, change ‘and 6.071 s’ to ‘, and 6.071 s’
Line 99, change ‘and the compound V’ to ‘which’
Line 108, change ‘elucidation of the toxin compound’ to ‘Elucidation of the Toxin Compound’
In Table 2, change ‘Absorption’ to ‘absorption’
Line 142 and Line 148, change ‘and’ to ‘, and’
Line 150, change ‘121.8 and’ to ‘121.8, and’
Line 152, change ‘159.8 and’ to ‘159.8, and’
Line 161, change ‘virulence of exogenous’ to ‘Virulence of Exogenous’
Line 179, change ‘structure and’ to ‘structure, and’
Line 183, change ‘, AG-3 PT subgroups’ to ‘, and AG-3 PT subgroups’
Line 184 and Line 189, change ‘and’ to ‘, and’
Line 195, change ‘Deutero- and’ to ‘Deutero-, and’
Line 201, Line 207, and Line 212, change ‘and’ to ‘, and’
Line 226, change ‘isolation and inoculation’ to ‘Isolation and Inoculation’
Lines 229-230, change ‘(Varieties: NC89, one of the 229 commonly cultivated susceptible variety).’ to ‘(Var.: NC89, one susceptible variety commonly cultivated).’
Line 234, change ‘toxin’ to ‘Toxin’
Line 236, change ‘in’ to ‘at’
Line 250, rewrite ‘repeat for three times.’
Line 268, change ‘grinded’ to ‘ground’
Line 281, change ‘pathogenicity of purified toxin compound’ to ‘Pathogenicity of Purified Toxin Compound’
Line 283, change ‘2 mg/mL, 4 mg/mL’ to ‘2 mg/mL, and 4 mg/mL’
References, 1) please provide the abbreviation of journal name; 2) Line 360, please provide the correct journal name; 3) The right description of title should be provided according to the guidelines of Journal of Fungi.
Author Response

(The authors gave the same response as above.)

Round 2
Reviewer 1 Report
no